# Combination of FDG PET/CT Radiomics and Clinical Parameters for Outcome Prediction in Patients with Hodgkin’s Lymphoma

**DOI:** 10.3390/cancers15072056

**Published:** 2023-03-30

**Authors:** Claudia Ortega, Yael Eshet, Anca Prica, Reut Anconina, Sarah Johnson, Danny Constantini, Sareh Keshavarzi, Roshini Kulanthaivelu, Ur Metser, Patrick Veit-Haibach

**Affiliations:** 1Joint Department of Medical Imaging, University Health Network-Mount Sinai Hospital, Women’s College Hospital, University Medical Imaging Toronto, University of Toronto, Toronto, ON M5T 1W7, Canada; 2Department of Diagnostic Imaging, Chaim Sheba Medical Center, Ramat Gan 5210000, Israel; 3Division of Medical Oncology and Hematology, Princess Margaret Cancer Centre, University Health Network, Toronto, ON M5G 2C4, Canada; 4Department of Medical Imaging, Belilinson Hospital, Rabin Medical Center, Tel Aviv University, Petah Tikva 4941492, Israel; 5Department of Diagnostic Imaging, William Osler Health System, 2100 Bovaird Avenue East, Brampton, ON L6R 3J7, Canada; 6Biostatistics Division, Dalla Lana School of Public Health, University of Toronto, Toronto, ON M5T 3M7, Canada

**Keywords:** hodgkin lymphoma, predictor models, radiomics, PET/CT

## Abstract

**Simple Summary:**

Hodgkin lymphoma is the most common malignancy of the lymphatic system, usually affecting young people and with favorable response to treatment. Tailored treatment is important in this patient population to achieve good therapy response while not exposing them to higher treatment related toxicity. These patients undergone PET/CT studies at baseline for staging as part of their workup. PET/CT is an hybrid modality including morphological and molecular images and most of the Hodgkin lymphomas are radiotracer avid. Radiomics is a new field analyzing features within the images itself which are not visible to the naked eye. Our hypothesis was that some radiomics features in the PET or CT modality when combined with clinical variables (laboratory, type of tumor, etc.) may help in prediction of how patients would respond to therapy, risk of relapse and if they might need to have additional therapies.

**Abstract:**

Purpose: The aim of the study is to evaluate the prognostic value of a joint evaluation of PET and CT radiomics combined with standard clinical parameters in patients with HL. Methods: Overall, 88 patients (42 female and 46 male) with a median age of 43.3 (range 21–85 years) were included. Textural analysis of the PET/CT images was performed using freely available software (LIFE X). 65 radiomic features (RF) were evaluated. Univariate and multivariate models were used to determine the value of clinical characteristics and FDG PET/CT radiomics in outcome prediction. In addition, a binary logistic regression model was used to determine potential predictors for radiotherapy treatment and odds ratios (OR), with 95% confidence intervals (CI) reported. Features relevant to survival outcomes were assessed using Cox proportional hazards to calculate hazard ratios with 95% CI. Results: albumin (*p* = 0.034) + ALP (*p* = 0.028) + CT radiomic feature GLRLM GLNU mean (*p* = 0.012) (Area under the curve (AUC): 95% CI (86.9; 100.0)—Brier score: 3.9, 95% CI (0.1; 7.8) remained significant independent predictors for PFS outcome. PET-SHAPE Sphericity (*p* = 0.033); CT grey-level zone length matrix with high gray-level zone emphasis (GLZLM SZHGE mean (*p* = 0.028)); PARAMS XSpatial Resampling (*p* = 0.0091) as well as hemoglobin results (*p* = 0.016) remained as independent factors in the final model for a binary outcome as predictors of the need for radiotherapy (AUC = 0.79). Conclusion: We evaluated the value of baseline clinical parameters as well as combined PET and CT radiomics in HL patients for survival and the prediction of the need for radiotherapy treatment. We found that different combinations of all three factors/features were independently predictive of the here evaluated endpoints.

## 1. Introduction

Hodgkin lymphoma (HL) is one of the most common malignancies involving the lymphatic system. The 5-year survival rate for all people with Hodgkin lymphoma is high, with an overall rate of around 87% [1].

Given the bimodal peak incidence with high rates of presentation at a young age, an individualized, risk-adapted therapy is desirable to maintain high cure rates while minimizing treatment-related toxicity [2,3].

Correct identification of predictive biomarkers that correlate with poor therapy response and an overall poor prognosis is essential for a personalized therapy approach, which is crucial to select patients that would benefit from an initial more aggressive therapy while avoiding overtreatment in patients with a high likelihood of a good prognosis [4,5,6,7].

Molecular imaging with clinical standard positron emission tomography (PET)/computed tomography (CT) using the radiopharmaceutical ^18^F-fluoro-deoxy-glucose (FDG) is the main imaging procedure for baseline staging of lymphoma, interim response assessment, and evaluation of residual disease in many jurisdictions worldwide [8].

Standardized uptake value (SUV) obtained from FDG PET/CT scans is the most widely used parameter for lesion depiction and characterization, and it provides a reliable assessment of tumor activity, tumor aggressiveness, and response to treatment [9].

However, SUV is not reflective of the underlying spatial distribution of tracer activity within a tumor itself, which can be particularly heterogeneous in lymphoma [10]. The unequal distribution of tracer activity within a tumor on FDG PET/CT is a manifestation of this ‘intra-tumor heterogeneity’, which can be measured by analyzing the variation in the spatial arrangements of voxel intensities [11].

In recent years, there has been increasing interest in radiomics, the science of extracting and analyzing quantitative and mineable features from standard-of-care biomedical images to create texture analysis of cross-sectional images (CT, MRI, and PET), which may provide detailed information of the underlying pathophysiology. Radiomics features of a tumor may provide additional information regarding tumoral biology and behavior [12,13,14,15,16].

Numerous studies have investigated intra-tumor heterogeneity on PET/CT in patients with brain, head, neck, thyroid, lung, breast, esophagus, pancreas, colon, and cervix neoplasms, as well as in patients with sarcomas and lymphomas [17,18,19,20,21].

Current clinical lymphoma biomarkers incorporate cellular and molecular data to classify specific disease subtypes and predict clinical behavior [22].

The association between intra-tumor image-based heterogeneity and biological heterogeneity has been shown to correlate with clinical outcomes such as treatment response and survival in a variety of tumor types, including lymphoma. This suggests that radiomic biomarkers can be developed and cross-referenced with established clinical cellular and molecular biomarkers to better predict outcomes and influence evidence-based clinical decision-making in patients with lymphoma [22,23,24,25,26,27,28,29,30].

This study aims to evaluate the prognostic value of joint PET and CT radiomics combined with standard clinical parameters in patients with HL. We hypothesize that some radiomic features within the baseline PET/CT may predict survival outcomes.

## 2. Materials and Methods

### 2.1. Study Cohort

In this institutional review board-approved retrospective study, 88 patients diagnosed and treated in a tertiary referral center with HL from September 2012 to June 2016 were evaluated. Given the retrospective nature of the analysis, consent was waived.

All patients had complete clinical records, including pathology reports from either nodal or extra-nodal biopsies, descriptions of sites of involvement, presence of bulky disease, Ann Arbor Stage, and B symptoms. Furthermore, all standard of care bloodwork, systemic treatment planned and received, as well as the provision of radiotherapy treatment along with response assessment for each line of therapy, were recorded. Follow-up times and progression-free survival were also registered.

Bulky disease was defined as more than 10 cm in any diameter.

Complete metabolic response (CMR) was defined as a disease at the end of therapy PET/CT below the Deauville score criteria of 4 [31,32,33,34,35,36,37].

### 2.2. Imaging Acquisition

^18^F FDG PET/CT was performed in these patients as a component of baseline staging. Images were obtained according to our institutional protocol, as follows: [23]. PET was performed on a Siemens mCT40 PET/CT scanner (Siemens Healthcare). Patients were positioned supine with their arms outside the region of interest. Images were obtained from the top of the skull to the upper thighs. Iodinated oral contrast material was administered for bowel opacification; no intravenous iodinated contrast material was used. Patients were asked to avoid exercise for 24 h and fast for 6 h before the examination. Patients received an IV injection of 5 MBq/kg (a range of 250–550 MBq) of FDG.

Overall, 5–9 bed positions were obtained, depending on patient height, with an acquisition time of 2–3 min per bed position. CT parameters were: 120 kV; 3.0 mm slice width; 2.0 mm collimation; 0.8 sec rotation time; 8.4 mm feed/rotation. A PET emission scan using time of flight with scatter correction was obtained, covering the identical transverse field of view. The PET parameters were as follows: image size: 2.6 pixels; slice: 3.27; and a 5-mm full width at half maximum (FWHM) Gaussian filter type.

### 2.3. Textural Analysis

Textural analysis of the PET/CT images was performed using the freely available software LIFE X (lifexsoft.org version 6.0 May 2020) via the quantitation of various radiomic features based on the spatial arrangement and variation of pixel intensities within a defined volume of interest [38]. The radiomic features were extracted from the segmented volumes in accordance with the image biomarker standardization initiative (IBSI) guidelines. Primary contour on FDG-avid nodal and extra-nodal lesions was performed semi-automatically by the software (with minor manual correction when needed) using a thresholding method to define each volume of interest (VOI) by two radiologists with >5 years of experience (YE and CO) and supervised by a senior radiologist with >10 years of experience (PVH).

PET volumes of interest (VOI) were defined based on (a) background thresholds, (b) peak thresholds, (c) thresholds at 40%, and (d) thresholds at 70% of the SUVmax PET VOI [26].

Individual lesions were measured, with a maximum of two per organ when multiple as per RECIST guidelines [39,40] and then labeled as nodal or extra-nodal involvement for each specific site (Figure 1). Lesions smaller than 64 voxels were excluded since they did not fulfill the minimum size criteria for feature extraction by the radiomics software.

Since a thresholding method is not available for the CT component, the contours for the CT-derived volume of interest were performed manually, slice-by-slice, to cover the entire tumor volume as previously described in the literature.

Sixty-five radiomic features (RF) were obtained by the software, including conventional metrics features reporting the mean, median, maximum, and minimum values of the voxel intensities on the image; size and shape histogram-based features such as volume, compacity, and sphericity, including their asymmetry (skewness), flatness (kurtosis), uniformity, and randomness; and textural features (such as GLCM (Gray-Level Co-occurrence Matrix), GLRLM (Grey-Level Run Length Matrix), NGLDM (Neighborhood Grey-Level Different Matrix), GLZLM (Grey-Level Zone Length Matrix).

### 2.4. Statistical Analysis

In this study, two main outcomes were considered. First, Progression Free Survival (PFS) is defined from the date of diagnosis to the date of first progression (or relapse) or date of death or last follow-up. Events are progression or death. A second endpoint named radiotherapy outcome is defined as the evaluation of whether radiomics at baseline PET can predict the need for radiotherapy after the completion of chemotherapy. The latter endpoint is a binary outcome, where those who received radiation are assigned a value of 1, while those who did not receive radiation are assigned a value of 0.

The characteristics of patients were presented as means and standard deviations for continuous variables and as frequencies and percentages for categorical variables. Univariate and multivariate models were used to determine the role of baseline demographics, clinical and laboratory characteristics, and FDG PET/CT radiomics in predicting the outcome of patients with lymphoma. A binary logistic regression model was used to determine potential risk factors for radiotherapy outcomes and the odds ratios (OR) and 95% confidence intervals (CI) were reported. The Cox proportional hazards regression, on the other hand, was used to determine PFS outcome factors and to calculate hazard ratios (HR) with 95% CI. The initial selection of informative variables for creating the best prediction models was accomplished through univariate analysis and repeated 10-fold cross-validation. Cross-validation was applied to all classes of baseline demographics, clinical, and FDG PET/CT radiomics variables to compensate for the lack of a validating cohort and to decrease the possibility of over-fitting the final model. In both logistic and Cox models, variables with a *p*-value of less than 0.10 in the univariate analysis were considered for inclusion in the multivariate analysis, and variables with a *p*-value of less than 0.05 were retained in the final model considering the backward elimination method. Pearson correlation is calculated to check the correlation between clinical, PET, and CT radiomic factors. In addition, predictors with high variance inflation factors are excluded from the models to avoid multicollinearity caused by correlated predictors. The average Brier score and the area under the receiver operating characteristic curve (AUC), which indicates the predictive accuracy of a model, were used to determine if the CT and PET variables would improve predictive accuracy over the demographic and clinical risk factors. All statistical analysis was performed in R (version 3.6.3, R Foundation for Statistical Computing, https://www.R-project.org/ May 2021).

## 3. Results

### 3.1. Study Population

Overall, 88 patients, 42 women (48%) and 46 men (52%), with a median age of 43.3 (range 21–85 years), were included.

Initial curative treatment was intended for all the patients. Combined doxorubicin + bleomycin + vinblastine + dacarbazine (ABVD) was the initial therapy of choice in 91% (*n* = 79) of patients, with 62% (*n* = 54/88) receiving 6 cycles and 94% of them (*n* = 82/88) completing therapy as initially planned at tumor boards. Of those, 84% (*n* = 72/88) achieved CMR.

Overall, 48% (*n* = 43/88) underwent additional radiotherapy for residual FDG-avid disease or due to initial bulky, disease achieving a complete metabolic response in 95% (*n* = 41/43).

At a median follow-up of 33.9 months (range 6–65), response to treatment was complete response (CR) in 88% of patients (*n* = 76), progressive disease (PD) in 8% of patients (*n* = 7), partial response (PR) in 2% of patients (*n* = 2), stable disease (SD) in 1% of patients (*n* = 1) and not evaluated in 2% of patients (*n* = 2) because of loss of follow-up. There were 10 adverse events during the follow-up period (defined as death, progression based on follow-up CT or PET/CT, or relapse), corresponding to 11.4%.

A summary of the patient population demographics, clinical information, and laboratory results is presented in Table 1 and Table 2, respectively.

### 3.2. Univariate Analysis

The statistically significant results of the univariable Cox regression analysis for CT, PET, and clinical parameters when considering either nodal-only involvement or all sites of disease involvement, as well as correlation with PFS and predictors of radiotherapy, are summarized in Table 3, Table 4 and Table 5, respectively. Of note, only one CT parameter (the GLZLM SZHGE mean) was found to be significant for the prediction of the need for radiotherapy in both categories (nodal vs. all sites), whereas several yet similar parameters (shape and GLRLM) were found to be significant for the prediction of the PFS endpoint. The results for PET showed similar trends: shape, GLRLM, and GLZLM features were found to be significant in all evaluation categories. Interestingly, a rather ‘standard’ feature such as TLG was found to be predictive as well. A summary of bivariate correlation coefficients is presented in Table 6.

The complete UVA results for all the clinical variables are included as Appendix A.

### 3.3. Multivariate Analysis (MVA) Parameters as Predictors of PFS

Multivariable Cox regression analysis was performed based on significant parameters (*p* < 0.1) from univariate analysis (UVA). MVA was performed in a backward manner with a stay criterion of *p* < 0.05. The parameters with the lowest *p*-value (clinical as well as imaging parameters) were used for model building for the PFS model, including Albumin (*p* = 0.034); ALP (*p* = 0.028), and CT grayscale parameter grey level run length matrix non-uniformity for a run (GLRLM GLNU mean (HR = 2.52, 95% CI (1.22, 5.18) *p* = 0.012)). Significant parameters and Forrest plots are presented in Table 7 and Table 8 and Figure 2 and Figure 3, respectively. Graph plot are presented in Figure 4 and Figure 5.

### 3.4. MVA Parameters Predictors of Need for Radiotherapy

For the prediction of the need for radiotherapy, a few parameters were significant in the UVA, including advanced stages (Stages III and IV combined). This parameter was however excluded from the MVA given that it was already predefined by the images and could potentially introduce bias. Therefore, first-order PET parameter SHAPE Sphericity (OR = 1.9, 95% CI (1.05, 3.42) *p* = 0.033); CT parameter grey level zone length matrix high gray-level zone emphasis (GLZLM SZHGE mean (OR = 2, 95% CI (1.08, 3.73), *p* = 0.028)); PARAMS XSpatial Resampling (OR = 2.1, 95% CI (1.2, 3.68), *p* = 0.0091); as well as abnormal hemoglobin results (OR + 0.26 (0.09, 0.78), *p* = 0.016) remain as independent features in the final model for the binary outcome as predictors of radiotherapy (AUC = 0.79).

## 4. Discussion

In our study, we evaluated the utility of combined PET/CT radiomic features as well as clinical parameters for outcome prediction in patients with Hodgkin’s lymphoma.

So far, only a few radiomics studies have been performed in HL populations addressing outcome prediction, and even fewer have considered clinical parameters as well as combined PET and CT features for outcome prediction. We found that CT as well as PET radiomics combined with clinical parameters might be able to help predict outcome endpoints such as PFS as well as the need for additional radiotherapy.

We found several radiomics parameters from baseline FDG PET/CT to be predictors of survival rate and predictors of the need for radiotherapy in the univariable analysis. However, when multivariable models were designed, considering the parameter with the lowest *p*-value for the model building, no PET-related parameter was found to be an independent predictor for PFS. This is concordant with a few earlier studies that evaluated a similar question, including first-order parameters such as SUVmax. For example, Frood et al. [27] recently published a meta-analysis of baseline PET/CT imaging parameters as a predictor of treatment outcome in Hodgkin and diffuse large B-cell lymphomas (DLBCL). In the meta-analysis, 10 studies assessing SUVmax as a predictor of response are included, however, none of the studies evaluated radiomics features. The largest study, by Akharti et al. [35], demonstrated that SUVmax could not be applied to predict either PFS or OS in 267 patients. Interestingly, in our study, a CT second-order parameter was a predictor of survival when combined with clinical parameters such as an abnormal albumin level and an elevated ALP.

Driessen, J. et al. [25] recently presented a radiomics analysis in a larger cohort of patients with relapsed HL. They found that a combination of radiomics and clinical features results in a strong prediction model for 3-year time to progression. The model uses robust PET features that address inter-lesional heterogeneity in the distance, metabolic volume, and SUV but did not include any second or higher-order radiomic features, as compared to our study. In addition, this investigation did not include a radiomics evaluation of the CT component of the PET/CT.

A recent study by Zhou and co-workers evaluated if the radiomic features of baseline FDG PET could predict the prognosis of Hodgkin lymphoma [36]. They found that long-zone high gray-level emphasis and Dmax were independently correlated with 2-year progression-free survival, although this study did not evaluate complementary CT-radiomics and did not integrate any clinical information into their AUC analysis. Furthermore, they evaluated a smaller number of patients, which were further divided into a training and validation data set, which likely decreased statistical robustness.

Another study has taken a somewhat different approach, evaluating 45 patients receiving R-CHOP (Rituximab+ Cyclophosphamide + Doxorubicin + Vincristine + Prednisone) chemotherapy for DLBCL evaluating the ability to predict therapy response [37]. Here, the authors concluded that SUV_max_ and gray-level co-occurrence matrix dissimilarity were independent predictors of lesions with an incomplete response.

Milgrom, S.A. et al. [29] analyzed a cohort of 251 mediastinal HL patents using another freely available software (IBEX). They found that the first-order parameters MTV and TLG are associated with disease progression in HL. None of the second-order parameters were predictors of progression in their cohort either.

Lue et al. [24] investigated 11 first-order and, 39 higher-order features in 42 patients with HL to predict PFS and OS. With 21 events in the cohort (12 relapses, 9 deaths), it was demonstrated that SUV, kurtosis, stage, and intensity non-uniformity (INU) derived from the grey-level run length matrix (GLRLM) were independent predictors of PFS, and only disease stage and INU derived from the GLRLM were independent predictors of OS.

Overall, compared to the relatively sparse, directly comparable literature, in our study, none of the PET-derived radiomic features were found to be independent features in the MVA for the PFS outcome. Since several PET-radiomic features were found to be significant in the UVA if we had evaluated only PET radiomic features, it might be that those parameters would have been significant in the MVA, and therefore, we would have more comparable results to other studies. However, in our investigation, PET radiomics parameters were ‘outperformed’ by the CT-radiomic features (which consequently ended up in the MVA) and were therefore not directly compared to the available studies. We feel that, since PET/CT is a hybrid imaging modality in clinical routine, both components (the PET and the CT) should be evaluated in a complementary fashion, and as demonstrated, there appears to be value in CT-derived textural features as well.

However, in our cohort, a PET first-order parameter, SHAPE Sphericity, and the CT second-order features, GLZLM SZHGE mean and PARAMS XSpatial Resampling, were independent predictors for the need for radiotherapy when combined with lower hemoglobin result at baseline lab work (AUC = 0.79) which again underlines the values for combined radiomic evaluation of PET and CT. It has to be pointed out that the clinical decision to apply additional radiotherapy is often multifactorial and that not only one clinical scenario indicates the need for radiotherapy in HL patients. In our institution, this decision is made mostly following the H10 trial [41], considering whether radiotherapy or a combined modality will be more beneficial for the early stages of disease in a personalized approach that is decided in most of the cases at multidisciplinary rounds by consensus. Individual factors such as the size of the radiation field and the organs included or in the vicinity, gender, age, and the risk of toxicity are all weighted factors in that decision.

Based on our analysis, the integration of combined CT and PET radiomics features might be of further guidance/help in deciding which patients might benefit from additional radiotherapy for the improvement of their disease outcome. Further studies have been addressing the same dilemma, including that of Picardi et al., who evaluated the correlation of histologically proven residual disease at the end of chemotherapy using PET/CT, showing a Deauville score of 4 foci after completion of the first line of chemotherapy [42].

Similar to other studies cited above, only first-order and morphologic PET radiomic features were found to be significant and, thus, not necessarily intrinsically related to voxel characteristics. For CT, however, two second-order features were found to be of value (i.e., GLZLM SZHGE). As for the comparative literature, no other studies evaluated predictors for the need for radiotherapy besides the bulkiness of the tumor, and therefore this finding may open a window for further analysis in larger cohorts.

Several other new studies have evaluated different aspects of radiomics, i.e., in MRI or PET, but those studies concentrated on the technical aspects of the analysis itself rather than the ability of radiomics for prediction. In addition, PET/CT radiomics has been thoroughly evaluated in non-Hodgkin lymphoma and in the context of prediction for bone marrow involvement, but as this was done for follicular lymphoma, these studies are not specifically relevant for HL patients [43,44,45,46].

Concerning the integration of clinical parameters, it has been shown in the literature that ALP is not necessarily a predictive clinical parameter on its own. While that is certainly valid from a dedicated clinical perspective, in our cohort it has been found to have predictive value in conjunction with the here evaluated imaging features. Thus, the integration of combined PET and CT radiomic features may elevate the value of specific clinical parameters when evaluated in conjunction.

## 5. Limitations

This study has several limitations. First, this is a retrospective analysis of data acquired in a single tertiary oncology center, so transferability to secondary centers might be limited. Second, the patient population is relatively small. Third, the cohort includes different types of HL; this non-uniformity may limit the translation of the results to patient’s clinical outcomes. Fourth, few adverse events were observed, as expected for this type of malignancy and population.

No cross-validation cohort analysis was performed, and although validation analysis cannot overcome the absence of a validation cohort, it can describe the variability in the findings and indicate the expected performance of the model in a distinct dataset.

Further prospective analysis to confirm our concepts and findings would be valuable.

## 6. Conclusions

We evaluated the value of baseline clinical parameters as well as combined PET and CT radiomics in HL patients for survival and prediction of the need for radiotherapy. We found that different combinations of all three factors/features were independently predictive of PFS outcome and radiotherapy outcome, as outlined above.

## Figures and Tables

**Figure 1 cancers-15-02056-f001:**
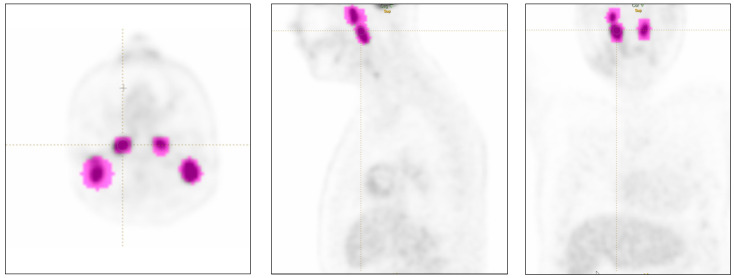
PET contouring using LifeX software of an 18-year-old male patient with Stage I Hodgkin Lymphoma.

**Figure 2 cancers-15-02056-f002:**
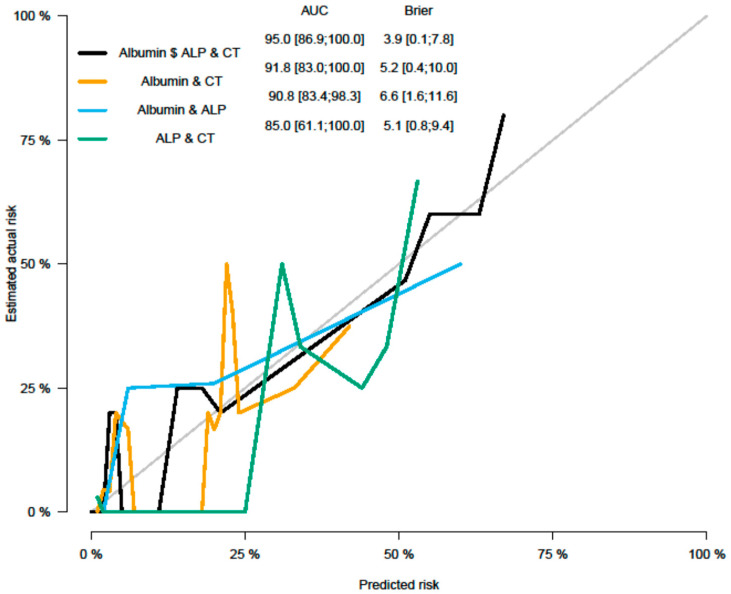
Calibration plots and AUC considering different variables in the model.

**Figure 3 cancers-15-02056-f003:**
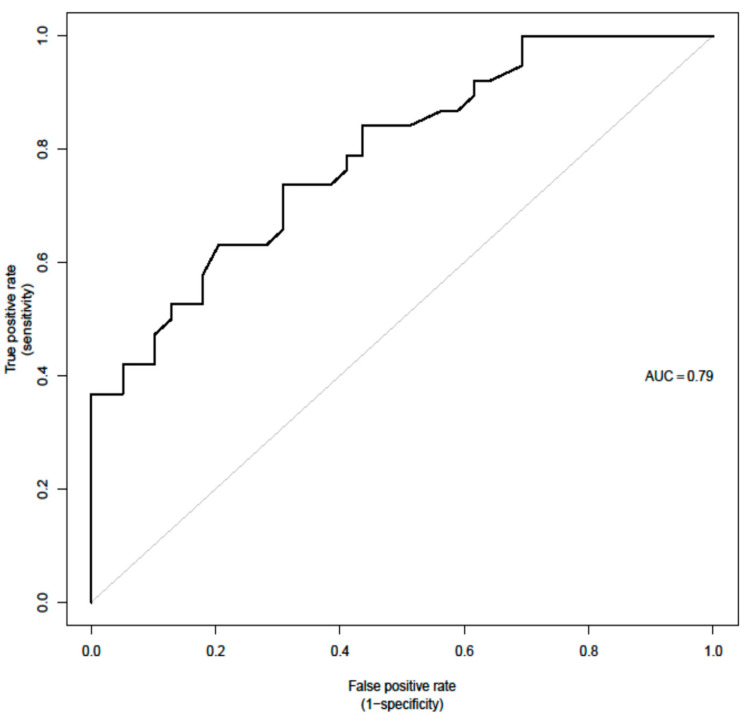
AUC for the final Radiotherapy model provided below.

**Figure 4 cancers-15-02056-f004:**
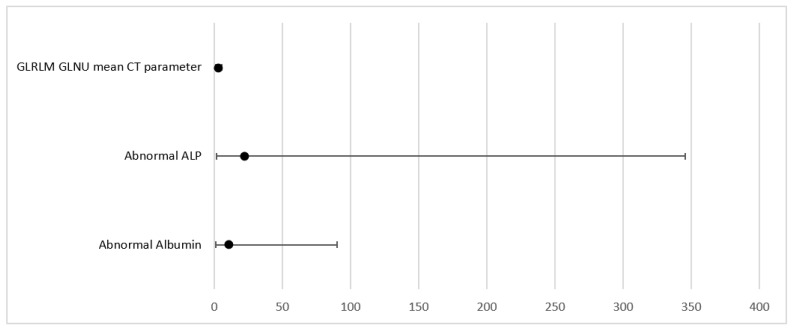
Forrest plot for MVA model correlating with PFS.

**Figure 5 cancers-15-02056-f005:**
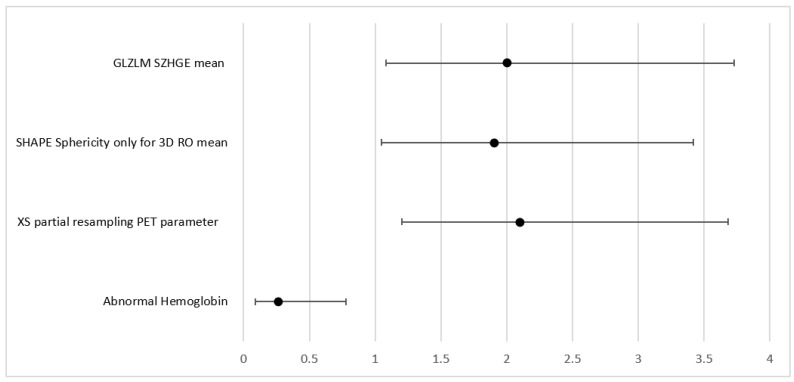
Forrest plot for MVA model when correlating with radiotherapy outcome.

**Table 1 cancers-15-02056-t001:** Summary of patient population characteristics.

Population	Age Years (Range)
88 patients	43.3 (21–85)
Distribution	*n* (%)
Female	42 (48)
Male	46 (52)
Pathology	*n* (%)
Classical Hodgkin’s lymphoma	16 (18)
Mixed cellularity classical Hodgkin lymphoma	5 (6)
Nodular lymphocyte predominant Hodgkin lymphoma	11 (12)
Nodular sclerosis classical Hodgkin lymphoma	56 (64)
Disease location	*n* (%)
Nodal disease	87 (98)
Extranodal disease	56 (63)
Bulky presentation	5 (5)
Overall stage	*n* (%)
Stage IA	5 (6)
Stage IIA	34 (39)
Stage IIB	8 (9)
Stage IIIA	14 (16)
Stage IIIB	4 (5)
Stage IVA	10 (11)
Stage IVB	13 (15)
Presence of B-symptoms	25 (28)
Chemotherapy regimen	*n* (%)
ABVE-PC	2 (2)
CT−A−AVD	1(1)
CT−A+AVD	4 (5)
LY−ABVD	79 (91)
OEPA/COPDAC	1 (1)
Missing info	1
Number of chemotherapy cycles	*n* (%)
1	1 (1)
2	8 (9)
3	1 (1)
4	19 (22)
5	1 (1)
6	54 (62)
7	3 (3)
Missing	1
Completed chemotherapy as planned	*n* (%)
No	4 (5)
Yes	83 (95)
Missing	1
Response to chemotherapy	*n* (%)
Complete response (CR)	72 (84)
Not evaluated (NE)	1 (1)
Progressive disease (PD)	5 (6)
Partial Response (PR)	7 (8)
Stable disease (SD)	1 (1)
Missing	2
Radiotherapy treatment	*n* (%)
Yes	42 (47)
No	46 (53)
Radiotherapy dose	Gy (range)
Mean	2869.6 (482.5)
Median	3000 (2000, 4000)
Response assessment after radiotherapy	*n* (%)
CR	41 (95)
PD	1 (2)
PR	0
ST	0

**Table 2 cancers-15-02056-t002:** Summary of Laboratory Parameters.

Parameter	Median (Range)	SD	Normal Range	Results	*n* (%)
Hemoglobin (g/L)	126 (79, 161)	16.8	120–160	Normal	42 (48)
			140–180	Abnormal	45 (52)
				Missing	11
WBC (×10^9^/L)	8.8 (1, 35.2)	5	4.0–11	Normal	54 (62)
				Abnormal	33 (38)
				Missing	11
Neutrophils (×10^9^/L)	6.8 (1.3, 31.8)	4.7	2.0–7.5	Normal	50 (57)
				Abnormal	37 (43)
				Missing	1
Lymphocytes	1.3 (0.2, 3.3)	0.6	1.5–4.0	Normal	41 (47)
				Abnormal	46 (53)
				Missing	1
Eosinophils	0.1 (0, 1.4)	0.2	0.04–0.4	Normal	62 (71)
				Abnormal	25 (29)
				Missing	11
ESR	33 (1, 115)	27.4	0–20	Normal	20 (27)
				Abnormal	55 (73)
				Missing	17
LDH	249 (133, 1542)	174	100–750	Normal	34 (40)
				Abnormal	52 (60)
				Missing	17
ALP	90 (45, 376)	56.6	40–150	Normal	75 (86)
				Abnormal	12 (14)
				Missing	1
ALT	17 (5, 147)	20.5	7–40	Normal	74 (85)
				Abnormal	13 (15)
				Missing	1
AST	18 (10, 252)	27.2	5–34	Normal	77 (92)
				Abnormal	7(8)
				Missing	4
Albumin	41 (26, 58)	5.3	32–53	Normal	53 (73)
				Abnormal	20 (27)
				Missing	15
Bilirubin	7 (1, 11)	11.7	0–22	Normal	76 (100)
				Abnormal	0
				Missing	12
Creatinine	69 (40, 123)	12.7	61–105	Normal	83 (95)
				Abnormal	4 (5)
				Missing	1
Calcium	2.4 (2.1, 2.7)	0.1	2.20–2.62	Normal	76 (93)
				Abnormal	6 (7)
				Missing	6

ESR: Erythrocyte sedimentation rate. LDH: Lactate dehydrogenase. ALP: Alkaline phosphatase.

**Table 3 cancers-15-02056-t003:** Univariate analysis (UVA) summary of statistically significant variables for CT parameters when correlated with PFS and radiotherapy prediction.

Parameter	HR	95% CI	*p* Value
CT parameters for NODAL involvement in PFS			
SHAPE Volume mL	1.99	1.22–3.26	0.0061
SHAPE Volume vx mean	1.97	1.16–3.34	0.012
GLRLM GLNU mean	2.06	1.25–3.4	0.0045
GLRLM RLNU mean	1.84	1.08–3.13	0.025
GLZLM GLNU mean	1.65	1.0–2.73	0.048
CT parameters for NODAL involvement to radiotherapy outcome			
GLZLM SZHGE mean	1.84	1.07–3.15	0.026
CT parameters for ALL SITES involvement to PFS			
SHAPE Volume mL mean	1.96	1.19–3.22	0.008
SHAPE Volume vx mean	1.97	1.17–3.33	0.011
GLRLM GLNU mean	1.99	1.24–3.19	0.0043
GLRLM RLNU mean	1.88	1.09–3.25	0.023
GLZLM GLNU mean	1.77	1.01–3.09	0.046
CT parameters for ALL SITES involvement to radiotherapy outcomes			
GLZLM SZHGE mean	1.67	1.01–2.76	0.047

**Table 4 cancers-15-02056-t004:** Univariate analysis (UVA) summary of statistically significant variables for PET parameters when correlated with PFS and radiotherapy prediction.

Parameter	HR	95% CI	*p* Value
PET volume 100% threshold for NODAL and PFS			
CONVENTIONAL TLG mean	1.89	1.15–3.12	0.013
SHAPE Volume mL mean	2.1	1.24–3.54	0.0055
GLRLM LRE mean	1.77	1.04–3.01	0.035
GLZLM LZE mean	1.66	1.15–2.4	0.0072
PET volume 100% threshold ALL SITES and PFS			
CONVENTIONAL TLGm mean	1.79	1.09–2.95	0.022
SHAPE Volume mean	1.96	1.16–3.32	0.012
GLZLM LZE mean	1.65	1.14–2.39	0.0085
NO predictors of radiotherapy outcome using PET volume 100% threshold were found			
PET volume 70% threshold for NODAL and PFS			
CONVENTIONAL TLG mean	1.94	1.15–3.27	0.013
SHAPE Volume mL mean	2.16	1.29–3.64	0.0037
GLRLM GLNU mean	2.1	1.08–4.09	0.028
GLZLM LZE mean	1.53	1.08–2.17	0.016
PET volume 70% threshold for NODAL and radiotherapy outcome			
HISTO Entropy log10 mean	0.6	0.36–0.99	0.047
PET volume 70% threshold ALL SITES and PFS			
CONVENTIONAL TLG mean	1.84	1.09–3.09	0.022
SHAPE Volume mL mean	2.02	1.2–3.42	0.0086
GLRLM GLNU mean	2.02	1.06–3.87	0.033
GLZLM LZE mean	1.53	1.08–2.17	0.018
PET volume 70% threshold ALL SITES and radiotherapy outcomes			
HISTO Entropy log10 mean	0.6	0.36–1.0	0.049
PET volume 40% threshold NODAL and PFS			
CONVENTIONAL TLG mean	1.53	1.03–2.28	0.036
SHAPE Volume mean	1.9	1.23–2.92	0.0037
GLRLM GLNU mean	2.27	1.3–3.96	0.0039
GLRLM RLNU mean	2.59	1.44–4.66	0.0015
PET volume 40% threshold ALL SITES and PFS			
CONVENTIONAL TLG mean	1.51	1.0–2.27	0.048
SHAPE Volume mean	1.87	1.2–2.91	0.006
GLRLM GLNU mean	1.79	1.09–2.95	0.022
GLRLM RLNU mean	1.71	1.09–2.68	0.019
GLZLM LZE mean	2.22	1.1–4.49	0.025
NO predictors of radiotherapy outcome using PET volume 40% threshold were found			

**Table 5 cancers-15-02056-t005:** Univariate analysis (UVA) summary of statistically significant variables for Clinical parameters when correlated with PFS and radiotherapy prediction.

Parameter	HR	95% CI	*p* Value
Effect of clinical variables on PFS			
Gender	2.15	0.56–8.3	0.27
Pathology subtype	7 × 10^−8^	0.00	0.84
Bulk disease	1.25	1.08–1.46	0.0035
Advanced stages (III + IV)	2.85	0.74–11.0	0.025
Presence of B symptoms	0.24	0.07–0.84	0.002
Chemotherapy completed as planned	0.11	0.02–0.54	<0.001
Abnormal Hemoglobin value	0.96	0.93–0.99	0.01
Abnormal neutrophils	1.06	0.96–1.17	0.025
Abnormal ALP	1.01	1–1.01	0.035
Abnormal Albumin	0.75	0.66–0.85	0.0023
Effect of clinical variables in radiotherapy outcome			
Advanced stages (III + IV)	0.05	0.02–0.14	<0.001
Presence of B symptoms	4.51	1.58–12.83	0.0048
Abnormal Hemoglobin	0.23	0.09–0.56	0.0012
Abnormal Lymphocytes	3.25	1.35–7.82	0.0086
Abnormal ESR	3.6	1.15–11.29	0.028
Abnormal ALP	0.18	0.04–0.85	0.031

**Table 6 cancers-15-02056-t006:** Bivariate correlation coefficients between variables. (*p*-value < 0.001 for all the below listed bivariate correlations).

Variable 1	Variable 2	Correlation Coefficients	*p*-Value
GLZLM_LZHGE_mean_pne	GLZLM_LZLGE_mean_pne	1	3.50 × 10^−261^
GLZLM_LZHGE_mean_pne	GLZLM_LZE_mean_pne	1	2.10 × 10^−223^
GLZLM_LZE_mean_pne	GLZLM_LZLGE_mean_pne	1	1.30 × 10^−219^
SHAPE_Volume__mL_mean_pne	SHAPE_Volume_VX_mean_pne	1	7.60 × 10^−166^
SHAPE_Volume_vx_mean_cne	GLRLM_RLNU_mean_cne	0.99	1.30 × 10^−70^
GLRLM_GLNU_mean_cne	SHAPE_Volume_vx_mean_cne	0.98	3.70 × 10^−60^
GLRLM_GLNU_mean_cne	GLRLM_RLNU_mean_cne	0.94	2.30 × 10^−43^
GLZLM_GLNU_mean_cne	GLRLM_RLNU_mean_cne	0.94	1.40 × 10^−36^
CONVENTIONAL_TLG_mean_pne	SHAPE_Volume__mL_mean_pne	0.91	6.10 × 10^−35^
CONVENTIONAL_TLG_mean_pne	SHAPE_Volume_VX_mean_pne	0.91	6.80 × 10^−35^
GLZLM_GLNU_mean_cne	SHAPE_Volume_vx_mean_cne	0.89	6.30 × 10^−28^
GLZLM_GLNU_mean_cne	GLRLM_GLNU_mean_cne	0.85	7.00 × 10^−24^
GLRLM_RLNU_mean_pne	GLRLM_GLNU_mean_pne	0.85	1.00 × 10^−23^
GLRLM_GLNU_mean_pne	SHAPE_Volume_VX_mean_pne	0.87	1.30 × 10^−23^
GLRLM_GLNU_mean_pne	SHAPE_Volume__mL_mean_pne	0.87	1.60 × 10^−23^
GLZLM_LZE_mean_pne	SHAPE_Volume__mL_mean_pne	0.83	2.30 × 10^−20^
GLZLM_LZE_mean_pne	SHAPE_Volume_VX_mean_pne	0.83	2.50 × 10^−20^
GLZLM_LZHGE_mean_pne	SHAPE_Volume__mL_mean_pne	0.83	2.60 × 10^−20^
GLZLM_LZLGE_mean_pne	SHAPE_Volume__mL_mean_pne	0.83	2.60 × 10^−20^
GLZLM_LZHGE_mean_pne	SHAPE_Volume_VX_mean_pne	0.83	2.90 × 10^−20^
GLZLM_LZLGE_mean_pne	SHAPE_Volume_VX_mean_pne	0.83	2.90 × 10^−20^
SHAPE_Volume__mL_mean_cne	SHAPE_Volume_VX_mean_pne	0.91	5.20 × 10^−19^
SHAPE_Volume__mL_mean_cne	SHAPE_Volume__mL_mean_pne	0.91	5.40 × 10^−19^
GLRLM_GLNU_mean_pne	CONVENTIONAL_TLG_mean_pne	0.76	6.40 × 10^−17^

**Table 7 cancers-15-02056-t007:** MVA model for PFS outcome.

Parameter	HR	(95% CI)	*p* Value
Abnormal Albumin	10.38	1.19–90.24	0.034
Abnormal ALP	21.89	1.39–345.3	0.028
GLRLM GLNU mean CT parameter	2.52	1.22–5.18	0.012

**Table 8 cancers-15-02056-t008:** MVA models for radiotherapy outcome.

Parameter	HR	(95% CI)	*p* Value
Abnormal Hemoglobin	0.26	0.09–0.78	0.016
XS partial resampling PET parameter	2.1	1.2–3.68	0.0091
SHAPE Sphericity only for 3D RO mean	1.9	1.05–3.42	0.033
GLZLM SZHGE mean	2	1.08–3.73	0.028

## Data Availability

Research data is available upon request to corresponding authors.

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
