# Peer review of "Combination of FDG PET/CT Radiomics and Clinical Parameters for Outcome Prediction in Patients with Hodgkin’s Lymphoma"

_cancers, 2023, doi:10.3390/cancers15072056_

Round 1

Reviewer 1 Report (New Reviewer)

Ortega et al submit a revised version of their manuscript willing to evaluate clinical and radiomics parameters for the outcome prediction in pts with Hodgkin’s Lymphoma. I have a couple of suggestions to further improve the manuscript.

General comments

1. The introduction is adequate to provide background information and the knowledge gap in the field.

2. The methods are described clearly.

3. All the tables have a dislocated format.

4. In table 1 the authors should include p-value comparing the distribution of their variables between subgroups (e.g. male vs female etc). If the effect of the subgroup is significant, the authors should adjust their univariate analysis accordingly. 

5. In table 4 is not clear if the univariate analysis reports crude or adjusted HR.

6. In the multivariate analysis, the authors should also report the AIC metric for model fitting. Also, given that the manuscript is optically poor with many confusing tables, the authors should consider presenting the MVA table as forrest plots.

Author Response

Dear Reviewer,

Thank you for reviewing our work. Please find enclosed the responses to your  suggestions.

Regards,

Claudia Ortega

Reviewer 2 Report (New Reviewer)

This trial evaluated a new imaging proposal. As it is not a standard technique, the authors should give more information regarding inter-observer and inter-equpment variability, the time to perform examinations, and so on. 

In Table 1-2, there is a relative high incidence of NLP-HL (about 12%).

In Table 1-2, there is an high incidence of extra-nodal involvement (about 63%), and a relative low incidence of advanced-stage disease (III-IV stages, about 47%).

The front-line treatment of patients should better specify in the Tables.

Author Response

Dear reviewer,

Thank you for taking the time to revise our study. Please find attached our answers to your valuable suggestions.

Regards,

Claudia Ortega.

Round 2

Reviewer 1 Report (New Reviewer)

The authors have carefully revised an already interesting manuscript. All my comments and concerns have been covered. I would like to congratulate the authors for their efforts.

Author Response

Dear reviewer,

Thank you for your valuable comments, much appreciated.

Best regards,

Claudia

Reviewer 2 Report (New Reviewer)

In the Result section, the authors should clearly state the rate of PFS at 33-months follow-up.

Thereare too many acronyms in the manuscript and the abstract (es, ALP, and so on..).

In thable 1, the sum of the Ann Arbor stages is 87 instead of 88. Is it correct?

The authors should improve the Discussion section and the references. Another approach in advanced-stage classic HL to improve decision making for subsequent radiotherapy or other anticancer treatment is combined end-of-treatment PET (PET-6) with imaging-guided core needle biopsy of FDG avid lymph nodes as suggested by Picardi et al. Please add in the references: Picardi M et al. European Journal of Cancer 2020; 132: 85-97).

Author Response

Dear reviewer,

Thank you for your comments.

Please find the answers below:

In the Result section, the authors should clearly state the rate of PFS at 33-months follow-up. Has been further clarified and added in the Results section. Thank you.

There are too many acronyms in the manuscript and the abstract (es, ALP, and so on..). Acronyms has been  revised throughout the manuscript, missing explanations has been added either between parenthesis or  at the end of the clinical and Lab data table.

In table 1, the sum of the Ann Arbor stages is 87 instead of 88. Is it correct?. You are correct, one case of Stage IIA was missing in the count. This has been corrected. Thank you.

The authors should improve the Discussion section and the references. Another approach in advanced-stage classic HL to improve decision making for subsequent radiotherapy or other anticancer treatment is combined end-of-treatment PET (PET-6) with imaging-guided core needle biopsy of FDG avid lymph nodes as suggested by Picardi et al. Please add in the references: Picardi M et al. European Journal of Cancer 2020; 132: 85-97). The need of radiotherapy paragraph has been rewritten. The mentioned article has been added as an alternative approach to validate the predictor value of end of therapy PET in this setting. Thank you for the valuable suggestion.

This manuscript is a resubmission of an earlier submission. The following is a list of the peer review reports and author responses from that submission.

Round 1

Reviewer 1 Report

The paper is very interesting and deals a very fascinating issue but not always easy to understand. The data analysis is complex, and it is surprising that radiomics features extracted form CT scan without iodinated contrast could be more significative than those coming from PET. As well as, several recognized prognostic factors in HL seems to be no more important in radiomics era, while others unusual, as ALP, have prognostic impact.

A clinical hematologist would raise some doubts and made some questions:

-        - Is it right to consider as main outcome the composite combination of Progression Free Survival and Overall Survival? Wouldn’t be enough to take only the PFS?

-        - Nodular lymphocyte predominant Hodgkin Lymphoma is a different disease from classical Hodgkin lymphoma. Most studies do not take both diagnoses together

-      - Although most patients were in early stage of disease, 62% of patients made 6 cycles of ABVD that is usually the treatment of advanced stages. How was the number of ABVD cycles decided?

-       - Most guideline for the treatment of HL recommend the need of radiotherapy after chemotherapy according to the initial stage and the number of scheduled cycles of ABVD, in the early stages. Was the indication to radiotherapy set only according the initial bulky disease and the final PET positive results?

-       - Was response to early PET, made after 2 cycles of ABVD, analyzed according to radiomics features at diagnosis?

-        - I’m not a statistician, so I ask whether is right that a cross validation analysis can overcome the absence of a validation cohort.

Minor comment:

-        - Several abbreviations are not explained in the text.

Reviewer 2 Report

GENERAL COMMENTS

The authors report on the prognostic significance of PET/CT radiomic features in Hodgkin lymphoma based on the analysis of 88 patients. The topic is interesting but the study suffers from several methodological issues. Reporting of the results is also suboptimal.

SPECIFIC COMMENTS

Major Comments

1.       The title is “Combination of FDG PET/CT Radiomics and Clinical Parameters for Outcome Prediction in Patients with Hodgkin’s Lymphoma”. However, clinical parameters are not adequately described in the manuscript. Laboratory covariates, which were significant in survival analysis, are not included in table 1. All clinical and laboratory parameters considered should be described in table 1 along with the appropriate cutoffs.

2.       More importantly, the impact of clinical parameters in univariate analysis is not described. A crucial question is: Do PET/CT radiomics add substantially to the prediction achieved by standard clinical parameters only?

3.       In table 1, it is shown that 56 patients had extranodal involvement but only 23 had stage IV. Please further clarify the extranodal involvement in earlier stages. Although it is acceptable, the percentage is unusually high!

4.       The combined endpoint PFS+OS is peculiar. In fact PFS includes both disease progression/relapse and death of any cause without prior treatment failure as events. Is PFS+OS equivalent to the conventional PFS definition? Please explain further and report how many events were recorded and if they were treatment failures or deaths of any cause without prior treatment failure.

5.       The endpoint “radiotherapy outcome” is not clinically sensible, because radiotherapy was administered either for PET+ residual disease or for bulky disease. There is no need to predict the need of radiotherapy for bulky disease, because this is predefined. Only radiotherapy for residual disease is relevant. Such an analysis should be restricted to patients with chemosensitive disease, as patients with primary progression of <PR are not typically irradiated but should not be classified in the favorable category of “non-irradiated” patients, given that they are chemoresistant.

6.       The authors perform multivariate survival analysis but do not present survival data, percentages and curves at all!!

7.       Correlations between clinical, CT and PET radiomics (by Pearson correlation coefficient) is described in “Methods” but not reported in the “Results”.

8.       Descriptive statistics for all the spectrum of radiomic parameters should be provided as a supplement.

Minor Comments

11. The definition of bulk, complete metabolic response etc should be moved from the “Results” to the “Methods” section.

22. I am not sure that patients with nodular lymphocyte predominant Hodgkin lymphoma should be analyzed together with those suffering from classical Hodgkin lymphoma.

33. Nodal disease was present in 87/88 patients. Which was the anatomic distribution of the single case of purely extranodal Hodgkin lymphoma? The complete absence of any nodal involvement in baseline PET/CT is extremely unusual in Hodgkin lymphoma!

44. Median follow-up would be more suitable than mean.

55. What kind of chemo did the patients not treated with ABVD receive?